# Novel Synergistic Approach for Bioactive Macromolecules: Evaluating the Efficacy of Goat Placenta Extract in PEGylated Liposomes and Microspicules for Chemotherapy-Induced Hair Loss

**DOI:** 10.3390/ph17081084

**Published:** 2024-08-19

**Authors:** Phitjira Sanguanboonyaphong, Phaijit Sritananuwat, Sureewan Duangjit, Anyamanee Lapmag, Watcharin Pumchan, Tanasait Ngawhirunpat, Praneet Opanasopit, Worranan Rangsimawong

**Affiliations:** 1Division of Pharmacy Practices, Faculty of Pharmaceutical Sciences, Ubon Ratchathani University, Ubon Ratchathani 34190, Thailand; phitjira.s@ubu.ac.th (P.S.); anyamanee.l@ubu.ac.th (A.L.); 2Innovation in Drug and Extract of Agriculture Research Group, Faculty of Pharmaceutical Sciences, Ubon Ratchathani University, Ubon Ratchathani 34190, Thailand; phaijit.s@ubu.ac.th (P.S.); sureewan.d@ubu.ac.th (S.D.); 3Division of Biopharmacy, Faculty of Pharmaceutical Sciences, Ubon Ratchathani University, Ubon Ratchathani 34190, Thailand; 4Division of Pharmaceutical Chemistry and Technology, Faculty of Pharmaceutical Sciences, Ubon Ratchathani University, Ubon Ratchathani 34190, Thailand; 5Somdet Phra Yuppharat Det Udom Hospital, Ubon Ratchathani 34160, Thailand; 6Department of Industrial Pharmacy, Faculty of Pharmacy, Silpakorn University, Nakhon Pathom 73000, Thailand; ngawhirunpat_t@su.ac.th (T.N.); opanasopit_p@su.ac.th (P.O.)

**Keywords:** goat placenta, bioactive compound, PEGylated liposomes, microspicules, chemotherapy-induced hair loss, hair growth

## Abstract

Chemotherapy-induced hair loss is a distressing side effect of cancer treatment, and medical interventions are often needed to address this problem. The objectives of this study were to evaluate the bioactivity of goat placenta (GP) extract on both normal and chemotherapy-induced hair cells and to develop PEGylated liposomes (PL) and microspicule (MS) formulations for promoting hair growth in patients with chemotherapy-induced hair loss. The bioactivities of GP extract on human follicle dermal papilla (HFDP) cells and cells damaged by chemotherapy were assessed. GP extract was incorporated into PLs and MS gel (PL-MS) and then investigated in vitro skin permeation and in vivo studies on the scalps of patients with chemotherapy-induced hair loss. GP extract stimulated HFDP cell proliferation in both normal and cisplatin-damaged cells. PL nanovesicles and MS gel worked synergistically to deliver macromolecular proteins into the skin and hair follicles. The application of GP extract-loaded PL-MS to the scalps of chemotherapy-treated patients for 12 weeks significantly enhanced the hair growth rate, without causing skin irritation. In conclusion, GP extract promoted the proliferation of hair cells damaged by chemotherapy, when this extract, combined with PL-MS, effectively delivered bioactive macromolecules across the skin and hair follicles, resulting in successful regrowth of hair post-chemotherapy.

## 1. Introduction

Chemotherapy, widely used in cancer treatment, often leads to various adverse effects, such as constipation, nausea, vomiting, diarrhea, fatigue, mucositis, and alopecia [1]. Hair loss, affecting up to 80–100% of patients treated with agents like cisplatin [2], doxorubicin, and docetaxel, results from the severe impact on rapidly proliferating organs like hair follicles, leading to extensive apoptosis of hair matrix keratinocytes. Although chemotherapy-induced hair loss is typically reversible, regrown hair often shows undesirable changes in color, texture, and growth rate, appearing grayish, rougher, slower-growing, and sparser [3,4]. These effects cause significant anxiety about physical appearance, self-confidence, and overall quality of life, even though the effects are often temporary [5].

Various therapeutic approaches, including scalp hypothermia, minoxidil, and calcitriol, aim to address alopecia. However, these agents are generally not recommended for post-cancer-treatment patients. Recent studies have highlighted the hair growth-promoting activity of human placenta, demonstrated by its ability to inhibit apoptosis and stimulate hair follicle proliferation in male C57BL/6 mice following chemotherapy [5,6]. The bioactive proteins and growth factors in human placenta and other mammalian placentas, such as goat placenta (GP), are similar [7,8]. GP contains numerous bioactive substances, including macromolecular proteins, growth factors, and amino acids [9]. Many exogenous growth factors serve as potential regenerative medicines for replacing or repairing damaged cells, tissues, and organs [10]. Additionally, placental extracts have shown promise in promoting hair regrowth by maintaining the anagen phase and facilitating cellular proliferation in hair follicles [6].

The targeted delivery of drugs to hair follicles is valuable for treating conditions like alopecia and acne [11]. However, skin’s structural and chemical barriers, as well as the specific targeting of hair follicles, require various penetration-enhancing techniques [12]. Liposomes show significant promise as nanocarriers for delivering therapeutic molecules to the skin and follicular region. Modifying liposomal surfaces with hydrophilic polymers enhances colloidal stability and increases skin permeation of numerous water-soluble compounds [13]. Microneedles, or microspicules (MSs), can enhance the skin permeability of macromolecular proteins painlessly [14]. Therefore, combining PEGylated liposomes (PLs) that entrap macromolecules from GP extract with minimally invasive MS may improve the local delivery of proteins and growth factors to target sites, offering an alternative treatment for hair regrow in chemotherapy-induced hair loss.

This study aimed to evaluate the bioactivity of GP extract on both normal hair cells and chemotherapy-induced cell damage and to develop PL and MS formulations for promoting hair growth in patients with chemotherapy-induced hair loss. We performed bioactivity studies of the GP extract on human follicle dermal papilla (HFDP) cells and developed a PL-MS formulation to enhance the skin permeability of bioactive macromolecules. We conducted in vitro permeation studies and visualized the permeation pathways to assess the formulation’s efficacy. Additionally, we performed an in vivo human study of GP extract-loaded PL-MS in patients with chemotherapy-induced hair loss.

## 2. Results

### 2.1. Bioactivity of the GP Extract on Hair Cells

Dermal papilla cells, which are isolated from the bulge of hair follicles, are expanded in culture to investigate their unique characteristics, such as aggregative behavior and the ability to induce new hair follicle formation [15]. As shown in Figure 1A, no cytotoxicity was observed in HFDP cells at any concentration of GP extract. The extract significantly enhanced HFDP cell proliferation at concentrations greater than 250 µg/mL (*p* < 0.05), thus indicating an increase in cell viability with increasing concentrations of GP extract. GP extracts significantly increased cell proliferation (Figure 1B) in a dose- and time-dependent manner (*p* < 0.05); moreover, compared with no treatment, treatment with 250–2000 µg/mL significantly promoted cell proliferation at 24 h, 48 h, and 72 h. These results suggest that GP extract effectively stimulates dermal papilla proliferation.

### 2.2. Effects of GP Extract on Chemotherapy-Induced Hair Cell Damage

As shown in Figure 2, 30 µg/mL of cisplatin significantly decreased cell viability to less than 50% after 24 h of incubation because of cell proliferation resulting from chemotherapy-induced hair-cell damage. After treatment with cisplatin for 48 and 72 h, the proliferation of HFDP cells strongly decreased. The cells pretreated with 2000 µg/mL GP extract exhibited significantly greater cell proliferation than did the cells treated with only cisplatin at all of the incubation times. In the case of cisplatin-induced HFDP cell damage, cells that underwent posttreatment with GP extract (2000 µg/mL) exhibited significantly greater proliferation than cells treated with cisplatin alone at 48 and 72 h. These findings suggested that GP extract can protect and repair HFDP cells and that both pre- and post-GP extract-treated cells increased the proliferation of cells in the chemotherapy-induced hair-cell damage.

### 2.3. Characterization of the GP Extract-Loaded PL–MS Gel

In a previous study, the physical properties of the GP extract included a brownish-red fibrous texture and the presence of water-soluble compounds, including high-molecular-weight proteins such as albumin, growth factors, and amino acids [9]. To improve the skin permeability of hydrophilic macromolecules, GP extract was loaded into PLs, which exhibited nanometer-scale vesicles (68.33 ± 0.30 nm (average diameter)), a narrow size distribution (polydispersity index: 0.31 ± 0.04), a negative zeta potential (−13.74 ± 0.91 mV), and a high %EE (57.09 ± 6.90%). As shown in Figure 3, the atomic-force microscopy (AFM) images of PLs exhibited morphology, including shape, structure, surface features, and size measurements. The spherical structure of the small unilamellar vesicles had a geometric mean diameter of 125.00 ± 14.03 nm and a height of 99.83 ± 12.30 nm.

The GP extract-loaded PL–MS gel was uniformly blended with an MS gel base, thus resulting in a formulation without coarse particles (Figure 4A). The higher viscosity inherent in gel-based substances contributed to increased stability, thus effectively suspending macromolecules, nanovesicles, and MS compared to the solution form. A microscopic examination demonstrated the needle-like structure of MS in the gel base, which featured tips emerging from two opposite ends, averaging 27.72 ± 3.40 µm in middle width and 429.61 ± 12.05 µm in spicule length (Figure 4B). This synergistic approach, which combines the nanoencapsulation technology of PL and the minimally invasive nature of MS, might be a successful method for bypassing the skin barrier and delivering biomacromolecules from GP extracts as transdermal and transfollicular delivery systems.

### 2.4. Skin Permeation and Deposition Study

The loading of GP extract into PLs improved the permeation of proteins from the extract through an artificial skin membrane (Strat-M^®^, Merck, Darmstadt, Germany) compared to the GP extract solution alone. However, the application of the MS formulation to the skin requires mechanical massage to insert the MS into the skin layer; thus, abdominal porcine skin was used as the skin membrane in this study. Mammalian skin contains endogenous proteins and peptides [16], which can influence the measurement of exogenous proteins and growth factors permeating the skin. This experiment used bovine serum albumin–fluorescein isothiocyanate conjugate (BSA-FITC) as a model macromolecular protein to assess the ability of the formulation to deliver hydrophilic macromolecules into and through the skin and hair follicles.

The skin permeation and deposition results are shown in Figure 5. The cumulative BSA-FITC permeation from the PL-MS was significantly greater than that from the solution across all of the sampling times. Moreover, compared with the solution, the PL demonstrated greater BSA-FITC permeation through the skin. PL enhanced the skin permeation of macromolecular protein more effectively than the gel, MS gel, and solution forms. However, no significant differences were observed among the formulations (Appendix A). The amount of BSA-FITC permeating through and depositing into the skin from the PL-MS was significantly greater than that from other formulations. Table 1 shows that the flux and K_p_ of BSA-FITC followed the order of PL-MS > PL > solution, thus indicating that PL-MS significantly enhanced the skin permeability of macromolecular proteins, representing a 14.72-fold increase compared to that of the solution. This observation suggested that the MSs improved the permeation of macromolecular proteins through and into the skin.

### 2.5. Confocal Laser Scanning Microscopy (CLSM) Study

To visualize the skin-penetration pathways of the PLs, BSA-FITC showed green fluorescence, and the rhodamine B 1,2-dihexadecanoylsn-glycero-3-phosphoethanolamine triethylammonium salt (Rh-PE)-probed nanocarriers showed red fluorescence, as shown in Figure 6. X-Y serial images of PLs with and without MS demonstrated the presence of fluorescent compounds at skin depths ranging from 0 µm to 170 μm. Furthermore, the red fluorescently probed PL nanovesicles were primarily located in the follicle openings, whereas PL-MS demonstrated nanovesicle distribution in both the follicle openings and the stratum corneum layer. This distribution facilitated the delivery of loaded BSA-FITC to the skin and follicular ducts, in which the creation of a penetration pathway effectively delivered nanovesicles and macromolecular proteins into and through the skin.

In addition, the fluorescence intensity of PL-MS was greater than that of PL at all skin-penetration depths. In the case of PL-MS, the fluorescence intensity of BSA-FITC-loaded nanocarriers was greatest at a skin depth of 60 µm, and that of Rh-PE-projected nanocarriers was greatest at a skin depth of 55 µm (Figure 7). This observation suggested that intact vesicles entered the stratum corneum, carrying vesicle-bound BSA-FITC into the skin. Moreover, the increased and deeper deposition of macromolecular proteins from the PL-MS than from the PL group indicates the effectiveness of the transdermal and transfollicular delivery system for macromolecular proteins.

### 2.6. In Vivo Human Study

Eleven females were included in the study. The mean age at diagnosis was 57.27 ± 8.90 years, and 63.63% of the patients had stage III breast cancer at diagnosis. In addition, the majority of patients (81.81%) received adjuvant chemotherapy consisting of doxorubicin and cyclophosphamide (AC regimen). For the safety endpoint, there were no allergic reactions, skin irritation, or redness, as evidenced by visual inspection (Figure 8). This indicates that the implementation of the GP extract-loaded PL-MS was safe. Hair-growth parameters, including length, density, and thickness, are illustrated in Figure 9. Treatment significantly improved hair growth, density, and shaft thickness at 4, 8, and 12 weeks compared to baseline (*p* < 0.05).

The hair growth rate on the treated side was significantly greater than that on the untreated side (*p* < 0.05). This formulation increased the hair growth rate, increasing the hair length by 12.32 ± 7.71 mm, 20.84 ± 10.39 mm, and 28.43 ± 8.55 mm at 4, 8, and 12 weeks, respectively. The hair growth rate of patients treated with the GP formulation was similar to that of healthy individuals. Generally, the healthy hair growth rate is 0.5 inches/month (12.5 mm/month) [17]. Therefore, GP extract-loaded PL-MS is vital for increasing the hair growth rate.

The mean hair density in the parietal areas of a normal human scalp is approximately 121.3 hairs·cm^−2^, whereas the mean hair diameter for individuals of Thai descent is approximately 85 µm [18]. Patients treated with chemotherapy always experience a reduction in hair density and thickness compared to healthy volunteers. However, there was a notable increase in hair density and thickness after applying the GP formulation for 12 weeks. The mean hair density was 89.5 hairs·cm^−2^, and the mean hair diameter was 64 µm, thus indicating a positive response to the treatment. Although the results were not different between the untreated group and the group treated with the GP formulation, the treatment did not negatively impact these parameters. Any potential benefits in terms of hair growth were not accompanied by significant changes in hair density or shaft thickness.

## 3. Discussion

The placenta is a complex organ that contains various bioactive compounds, such as amino acids, peptides, growth factors, enzymes, polydeoxyribonucleotides, vitamins, and trace elements. These compounds have been shown to alleviate fatigue, promote wound healing, and stimulate the regrowth of hair [6]. The human placenta stimulates hair growth through the Wnt/β-catenin pathway, which has been shown to improve hair growth in mice [19]. GP extract shares similarities with human placenta and contains many bioactive compounds, such as macromolecular proteins and various growth factors, including epidermal growth factor (EGF), fibroblast growth factor-2 (FGF-2), insulin-like growth factor-1 (IGF-1), and transforming growth factor beta-1 (TGF-β1), which play crucial roles in promoting hair growth and rejuvenation. In a previous study, EGF was shown to stimulate cell cycle progression and proliferation in dermal papilla cells [20]. FGF and IGF-1 promote hair follicle growth by inducing or maintaining the anagen phase and preventing cell death during the catagen phase [21,22]. The TGF-β family plays a key role in hair follicle development and cycling by enhancing the proliferation of HFDP cells and enhancing hair growth [23]. Overall, GP extract, which is rich in growth factors and other bioactive compounds, is essential for stimulating hair growth and supporting hair rejuvenation by enhancing dermal papilla cell proliferation and promoting healthy hair follicle function.

Cisplatin is a chemotherapy drug that is known to frequently cause the death of mechanosensory hair cells due to oxidative stress, DNA damage, and the release of inflammatory cytokines. The transcription factor STAT1, which is an important mediator of cell death, plays a role in regulating these processes across various cell types [2]. The sensitivity to cisplatin treatment varies depending on the concentration, duration of exposure, and the phase of the cell cycle. Cells exposed to cisplatin during mitosis are hypersensitive compared to those in interphase [24]. Moreover, cisplatin has been reported to induce HaCaT cell arrest and apoptosis within 24 h, resulting in the highest reduction in proliferation observed at this time point. These resistant cells also become less susceptible to the cytotoxic effects of cisplatin [25]. Therefore, lower proliferation was observed at 48–72 h, likely due to the remaining cells being in an arrest phase.

Chemotherapy drugs also increase fragile hair counts and miniaturize hair follicles due to damage related to apoptosis [26]. The human placenta has been reported to inhibit apoptosis and promote hair growth by increasing keratinocyte growth factor (KGF) expression and AKT phosphorylation [5]. KGF, which is part of the heparin-binding fibroblast growth factor family (FGF-7), supports the transition of hair follicles into the active growth phase (anagen) and encourages hair follicle proliferation [27,28]. The ability of GP extract to enhance HFDP cell proliferation, even when exposed to chemotherapy-induced damage, suggests that it may play a beneficial role in promoting regrowth of hair and reducing chemotherapy-induced hair cell damage. By providing essential nutrients and growth factors, GP extract can aid in repairing and regenerating hair cells, thus ultimately improving hair health and potentially alleviating chemotherapy-induced hair loss.

Hydrophilic macromolecules, which include proteins, growth factors, and amino acids, encounter limitations in passive skin penetration. The skin’s outermost layer (the stratum corneum) forms a lipophilic barrier that selectively allows for the passive penetration of small, potent, and moderately lipophilic molecules into the deeper skin layers [29]. Nanocarrier-loaded macromolecular proteins were formulated to effectively deliver proteins and growth factors across the skin. Surface-modified liposomes have proven to be effective in encapsulating biomacromolecules, such as amino acids, peptides, and proteins, from deer-antler velvets, thus ensuring their efficient delivery and safety [30]. The nanometer size of PL indicates that the sonication energy used in lipid suspensions can disrupt multilamellar vesicles, resulting in the formation of smaller unilamellar vesicles [31,32]. However, the diameter measured by AFM was higher than that size measured by a dynamic light-scattering particle size analyzer due to the flattening of vesicles. The shape of the liposomes can change after deposition on mica support, with deformation induced by the interaction between the vesicle and the support, as well as the continuous movement of the AFM tip, which depends on the vesicle composition [33].

The presence of PL vesicles enhances the ability of entrapped hydrophilic substances to permeate the skin. The liposomal structure creates a protective environment for hydrophilic substances, with the membrane bilayer becoming more polar in the presence of PEG–lipids, thus leading to increased efficiency in incorporating hydrophilic compounds. Additionally, PLs incorporating an edge activator (polysorbate 20) exhibit deformable vesicles, thus allowing for intact vesicles to enter the stratum corneum and carry vesicle-bound drugs into the skin [16]. The PEG chains on the liposome surface also interact with lipids in the stratum corneum, thus temporarily disrupting barrier function. This interaction enhances the penetration of liposomal formulations and facilitates the permeation of hydrophilic substances [34]. Although PLs play a crucial role as nanovesicles for delivering hydrophilic macromolecules through and into the skin, lipid nanocarriers have limited efficacy in deeply penetrating the skin due to alterations in the lipid structure of the stratum corneum [16]. Therefore, a combination of nanovesicle and microneedling techniques was developed to increase the ability of bioactive macromolecules permeated through the skin.

The needle-like structure of MSs, with tips extending from opposite ends, enables them to navigate through the stratum corneum barrier and the viable epidermis. The insertion of MSs into the skin created an efficient pathway for delivering proteins and growth factors. This approach bypasses the tightly packed stratum corneum and enhances the transport of macromolecules into the skin. Additionally, MSs can function as a microdermabrasion tool, thus removing the top layer of the skin barrier and further improving the skin permeability of compounds [35,36]. Consequently, the combination of MSs and the PL formulation, which has the potential to deliver macromolecular proteins into and through the skin, was selected for subsequent study.

The visualization of skin penetration pathways was conducted by using CLSM, which provides semiquantitative data on the distribution of fluorophores across different skin layers, including the stratum corneum, epidermis, dermis, outer and inner root sheath, cuticular area, and hair shaft [37]. Although large and hydrophilic molecules in solution face limitations in permeating the lipid matrix of the stratum corneum and penetrating into deeper skin layers, they can be distributed within corneocytes through the transcellular pathway and transported through hair-follicle orifices via the transfollicular pathway [11]. PL nanocarriers were designed to improve the skin permeability of hydrophilic macromolecules. Once PL vesicles are absorbed into the outermost layer of the skin, the entrapped compound is released from the vesicles and partitioned into the skin layers [16]. In the case of MSs, macromolecular proteins are delivered into the skin through two mechanisms: (1) MSs utilize minimally invasive techniques for the creation of micropores in the epidermis; and (2) MSs resurface the skin by removing the stratum corneum layer [29]. Thus, PL-MS is a potential transdermal and transfollicular delivery system for accessing biomacromolecules from GP extracts.

In cases of chemotherapy-induced hair loss, patients typically experience significant shedding, a reduction in hair density and shaft thickness, and gray hair, often leading to psychological distress and reduced quality of life. Although chemotherapy-induced alopecia is considered to be temporary, the recovery process requires time to reach the normal stage of hair development. Some patients report persistent alopecia for a period from 6 months to 3 years after chemotherapy [38]. The application of GP extract in the form of PL-MS may help to mitigate these adverse effects by promoting hair regrowth and maintaining regular hair growth. This approach provides a promising avenue for addressing chemotherapy-induced alopecia and improving the overall well-being of patients undergoing chemotherapy treatment.

Bioactive extracts from GPs contain numerous proteins, growth factors, and amino acids [9], which play essential roles in developing hair follicles and cycling [28]. Placenta extract from humans has demonstrated remarkable antiapoptotic and hair growth-enhancing effects on cyclophosphamide (chemotherapy drug)-induced alopecia in a mouse model by restoring the size of hair follicles and aiding in the recovery of hair growth. Additionally, the treatment repaired broken hair shafts that had not emerged through the epidermis, thus supporting the restoration of healthy hair growth and appearance [5]. FGF-2 successfully stimulates hair growth by promoting dermal papilla cell proliferation and increasing the size of hair follicles; moreover, when it is encapsulated in liposomes and hydrogels from silk fibroin, it can quickly regenerate hair and restore hair follicles to the anagen phase, thus providing a promising strategy for preventing hair loss in patients with alopecia areata [39]. Therefore, GP extract plays a crucial role in proliferating dermal papilla cells in hair follicles and preventing or repairing hair cell damage caused by chemotherapy. When delivered through a transdermal and transfollicular system using a PL-MS formulation, GP extract offers effective hair growth stimulation in patients with chemotherapy-induced hair loss. Although the current study provides valuable insights into the safety and efficacy of GP extract-loaded PL-MS, it is limited by its small sample size. Larger, more comprehensive studies involving a greater number of human volunteers are planned for future experiments to better evaluate the benefits and effects of this treatment.

## 4. Materials and Methods

### 4.1. Materials

GP extract was obtained from the Faculty of Pharmaceutical Sciences, Ubon Ratchathani University, Thailand [9]. HFDP cells and HFDPC growth medium (all-in-one ready-to-use) were purchased from Cell Applications, Inc., San Diego, CA, USA. 3-(4,5-dimethylthiazol-2-yl)-2,5-diphenyl tetrazolium bromide (MTT), cisplatin, and BSA-FITC (albumin (MW: 66 kDa) and FITC (MW: 389.4 Da)) were obtained from Sigma–Aldrich Co., St. Louis, MO, USA. Penicillin–streptomycin was purchased from Gibco BRL, Rockville, MD, USA. Phosphatidylcholine (Phospholipon^®^ 90 G), and N-(carbonyl-methoxypolyethylene glycol 2000)-1,2-distearoyl-sn-glycero-3-phosphoethanolamine sodium salt (PEG2000-DSPE) were obtained from Lipoid GmbH, Ludwigshafen, Germany. Cholesterol was obtained from Carlo Erba Reagent, Ronado, Italy. Polysorbate 20 (Tween 20) was obtained from Namsiang Group in Bangkok, Thailand. Rh-PE (Lissamine™) was purchased from Invitrogen, Carlsbad, CA, USA. Hyaluronic acid was purchased from P.C. Drug, Bangkok, Thailand. Ethylenediaminetetraacetic acid, disodium salt (EDTA-2Na), glycerin, and microcare PHC (phenoxyethanol, chlorphenesin and glycerin) were purchased from Krungthepchemi, Bangkok, Thailand. MSs extracted from *Spongilla lacustris* (98% spicules) were purchased from Hunan Sunshine Bio-Tech Co., Ltd., Changsha, China.

### 4.2. Bioactivity Study of GP Extract on HFDP Cells

HFDP cells were cultured in HFDPC growth medium supplemented with penicillin–streptomycin (1% *v*/*v*) under 95% CO_2_ in a humidified atmosphere at 37 °C. For the cytotoxicity evaluation, HFDP cells (1 × 10^4^ cells/well) in a 96-well plate were incubated overnight. After removing the medium, the cells were treated with GP extract at concentrations ranging from 1 to 4000 µg/mL for 24 h. Cell viability was assessed by using the MTT assay. In brief, after discarding the medium, the cells were washed with phosphate-buffered saline (PBS) at pH 7.4 and then incubated with MTT solution (final concentration of 0.5 mg/mL) for 3 h. Following the formation of formazan crystals by living cells, a volume of 100 μL of dimethyl sulfoxide (DMSO) per well was used to dissolve the crystals. The absorbance was determined at 550 nm by using a microplate reader (VICTOR Nivo^TM^ Multimode Plate Reader, PerkinElmer Life and Analytical Sciences, Inc., Llantrisant, UK).

The HFDP cell proliferation was evaluated by seeding 5 × 10^3^ cells into each well of a 96-well plate. After overnight incubation, different concentrations of GP extract were added to the cells for 24, 48, and 72 h. The proliferation of the cells treated with the GP extracts was measured by using an MTT assay and calculated by using Equation (1):(1)$$\%\;Cell\;viability\;or\;proliferation=\frac{Absorbance\;of\;treated\;cells}{Absorbance\;of\;untreated\;cells}\times 100$$

### 4.3. Cell Proliferation Following Chemotherapy-Induced Hair Cell Damage

Cisplatin (30 µg/mL) was used as a chemotherapy drug to treat cultured cells. In this study, HFDP cells (5 × 10^3^ cells/well) in a 96-well plate were exposed to cisplatin under either pretreatment or posttreatment conditions. For pretreatment conditions, HFDP cells pretreated with GP extract were exposed to the extract for 24 h before cisplatin was added for an additional 24 h. Cell proliferation was assessed by using the MTT assay at 24, 48, and 72 h, and the results were calculated by using the previous Equation (1).

For posttreatment conditions, the cells were initially treated with cisplatin for 24 h, after which GP extract was added. Cell proliferation was evaluated as the method described above.

### 4.4. Preparation of the PL-Loading GP Extract

The PLs were formulated with a controlled amount of phosphatidylcholine, cholesterol, and PEG2000-DSPE at a molar ratio of 10:2:0.12 mM, using thin film hydration and sonication methods. In brief, lipids were dissolved in a chloroform/methanol mixture (2:1, *v*/*v*) and thoroughly mixed in a round-bottom flask. The solvent evaporation was performed by using rotary evaporator, after which the lipid film was placed into a desiccator for approximately 6 h to ensure complete drying. GP extract (0.2% *w*/*v*) dissolved in PBS (pH 7.4), and 2.0% *w*/*v* polysorbate 20 were added to the lipid film to hydrate the liposome vesicles. The dispersion was then subjected to probe sonication in an ice bath for 30 min to reduce the size of the liposomal vesicles. Excess lipid components were removed via centrifugation, and the resulting supernatant was stored at 4 °C for later use.

The particle size, size distribution, and surface charge were measured by using a dynamic light-scattering particle size analyzer (Zetasizer Nano-ZS, Malvern Instruments, Worcestershire, UK), following the methods of Rangsimawong et al. (2024) [30]. For the determination of entrapment efficiency, 2 mL of the formulation was poured into a centrifugal filter tube (Amicon^®^ 100 K, Merck Millipore Ltd., Cork, Ireland.). Centrifugation was operated at 4000 rpm and 4 °C for 15 min. The nanocarriers entrapping the GP extract were placed into the filter device and subsequently combined with 0.1% Triton X-100 at a 1:1 ratio. The protein content of the collected nanocarriers was then measured by using a bicinchoninic acid (BCA) protein assay kit (Novagen^®^, EMD Millipore, Madison, WI, USA), following the manufacturer’s protocol. The percent of entrapment efficiency (%EE) was calculated using Equation (2).
(2)$$\%EE=\frac{Extracted\;protein\;from\;GP\;extract\;entrapped\;nanocarriers}{Initial\;extracted\;protein\;from\;GP\;extract\;loaded}\times 100$$

The morphology of PL vesicles was observed by AFM (SPA400, SPI4000, SII Seiko Nanotechnology, Tokyo, Japan) operating in tapping mode at 25 °C. Nanovesicles were diluted with distilled water before being dropped onto a mica sheet. The cantilever tip had a resonant frequency of ±10% at 210 kHz and a force constant of ±20% at 6.1 N/m (NT–MDT, Moscow, Russia). The scanning area was either 1 µm^2^ or 2.5 µm^2^, and the scan rate was approximately 1.0 Hz. AFM tapping-mode data were analyzed using SPI4000 version 4.17E (SII Seiko Nanotechnology, Tokyo, Japan).

### 4.5. Preparation of MS Gel Base for Loading PLs

MS gel base containing hyaluronic acid (1.5% *w*/*w*), glycerin (4.0% *w*/*w*), Microcare PHC (1.0% *w*/*w*), and EDTA-2Na (0.05% *w*/*w*) was mixed with MS (0.5% *w*/*w*). Afterwards, the GP extract-loaded PLs were added to the gel. The visual appearance was observed. The shape and size of the MSs were imaged under a light microscope (Nikon DS-Ri2, Nikon Corporation, Tokyo, Japan).

### 4.6. In Vitro Skin Permeation Study

Each piece of abdominal skin was collected from intrapartum stillborn piglets at a local farm in Sisaket Province, Thailand, with appropriate ethical approval for collection by the Investigational Review Board (05/2565/IACUC, Animal Experimentation Ethics Committee, Ubon Ratchathani University). The subcutaneous layer was carefully excised by using surgical scissors. The skin (with a 0.6–0.7 mm thick) was kept frozen at −20 °C. Before use, the samples were thawed in PBS (pH 7.4) at room temperature.

Typically, endogenous proteins and peptides in the skin affect the accurate measurement of exogenous proteins and growth factors permeating into the skin. To evaluate permeation in this experiment, BSA-FITC served as a model macromolecular protein, and its skin permeation was assessed by using vertical Franz-type diffusion cells. The receiver medium consisted of PBS (pH 7.4), which was continuously stirred by using a magnetic stirrer and maintained at a stable temperature of 32 ± 2 °C. The 0.5 mL formulation was applied onto the skin and gently massaged for 2 min (approximately 160 rubbing motions over a 2.01 cm^2^ skin area) by using a forefinger wearing a medical glove. Fluorescence analysis was conducted by collecting 500 µL samples of the receiver medium at 1, 2, 4, 6, and 8 h. The receiver compartment was replenished with an equivalent volume of PBS to maintain a constant volume throughout the experiment. Each sample was measured three times.

Fick’s law of diffusion was employed as a mathematical model to determine the skin permeation parameters. The cumulative permeation amount (μg·cm^−2^) over time (h) was plotted on a graph, and the steady-state flux was calculated from the slope of the linear region for each formulation. The permeability coefficient (K_p_) and enhancement ratio (ER) were calculated by using Equation (3) and Equation (4), respectively.
(3)$$K_{p}=\frac{Steady-state\;flux}{Donor\;concentration\;of\;the\;formulation}$$
(4)$$ER=\frac{Flux\;of\;skin\;penetration\;enhancing\;system}{Flux\;of\;solution\;form}$$

Following the 8-hour skin-permeation study, treated skins were rinsed with PBS (pH 7.4) to eliminate excess formulation from the skin surface. Subsequently, the treated skin was dissected into small pieces and extracted with 2 mL of PBS (pH 7.4), followed by sonication in an ultrasonic bath for 30 min. Fluorescence analysis was employed to analyze the quantity of extracted protein (BSA-FITC) deposited throughout the entire skin.

For the analysis of the amount of macromolecular protein (BSA-FITC), the samples were transferred to a 96-well plate (black). The fluorescence intensities of the compounds were determined in triplicate by using a fluorescence spectrophotometer (CLARIOstar^®^ Plus, BMG LABTECH, Ortenberg, Germany), with excitation at 485 nm and emission at 535 nm.

### 4.7. CLSM Study

After 8 h of in vitro skin permeation, the permeation pathway was visualized by using CLSM images of green fluorescent BSA-FITC as a model macromolecular protein and red fluorescent Rh-PE as a nanocarrier probe. Each treated skin sample was rinsed with PBS (pH 7.4) to eliminate excess formulation. The entire skin was immersed in an ample amount of methyl salicylate, and the skin surface and the permeation depth were visualized by using a CLSM (inverted Zeiss LSM 800 microscope, Carl Zeiss, Jena, Germany) instrument equipped with diode lasers (405, 488, and 561 nm). The images were captured by using a ×10 objective lens. ZEISS ZEN software version 2.3 was used to determine the fluorescence intensities at the middle horizontal line of images. The mean fluorescence intensity versus the skin depth of each formulation was plotted.

### 4.8. In Vivo Human Study

This prospective study was conducted in breast-cancer patients who experienced hair loss after completing chemotherapy at Somdet Phra Yuppharat Det Udom Hospital, Ubon Ratchathani, Thailand, between 1 August 2023 and 30 December 2023. This study was approved by the Ubon Ratchathani Provincial Health Office (Ethics No. SSJ.UB2566-043). Patients were eligible for this study if they were 18 years or older, had completed a chemotherapy regimen that included either AC (doxorubicin and cyclophosphamide) or FAC (5-fluorouracil, doxorubicin, and cyclophosphamide), and had an Eastern Cooperative Oncology Group (ECOG) performance status of ≤2 or a Karnofsky Performance Status (KPS) of ≥ 60%. Exclusion criteria included a history of allergic reactions to the extract or product ingredients, and the use of other hair growth-stimulation products.

Patients who met all of the eligibility criteria were required to refrain from applying any moisturizing products for at least 12 h before participating in the study and throughout the study period. A 3 cm × 3 cm area was marked on the patient’s scalp in the parietal area, and an appropriate amount of formulation was applied to the marked area twice daily. Hair growth rates were measured by using a digital Vernier caliper at baseline (Day 0) and at 4, 8, and 12 weeks. Changes in hair appearance, including the density and thickness of hair shafts, were observed by using API 202 for hair analysis (Aram Huvis Co., Ltd., Gyeonggi-do, Republic of Korea).

The primary outcomes were to compare the efficacy of the GP extract formulation between the applied site and untreated site at 4, 8, and 12 weeks compared to that at baseline. The efficacy outcomes included hair growth, density, and shaft thickness. Furthermore, the hair growth parameters included length, density, and thickness. This study observed the following safety outcomes: skin irritation, redness, and allergic reactions.

### 4.9. Data Analysis

The data are reported as the mean ± S.D. Statistical significance was assessed by using one-way ANOVA and Tukey’s post hoc test. The significance level was established at *p* < 0.05. For the in vivo study, the baseline characteristics were analyzed by using descriptive statistics. The primary outcomes were analyzed by using the Wilcoxon signed rank test. Statistical significance was based on a two-sided test with a significance level of *p* < 0.05. SPSS version 29 was used for the statistical analyses and data management.

## 5. Conclusions

Bioactive GP extract effectively stimulates the proliferation of dermal papilla cells and helps prevent or repair hair cell damage caused by chemotherapy. The GP extract-loaded PL–MS formulation demonstrated suitable physicochemical properties and facilitated the delivery of macromolecular proteins into the skin and hair follicles through transdermal and transfollicular pathways. The treatment of human scalp skin in patients experiencing chemotherapy-induced hair loss with a formulation containing GP extract promoted hair growth without causing skin irritation. Thus, GP extract-loaded PL-MS is a promising and effective formulation for promoting hair growth in chemotherapy patients.

## Figures and Tables

**Figure 1 pharmaceuticals-17-01084-f001:**
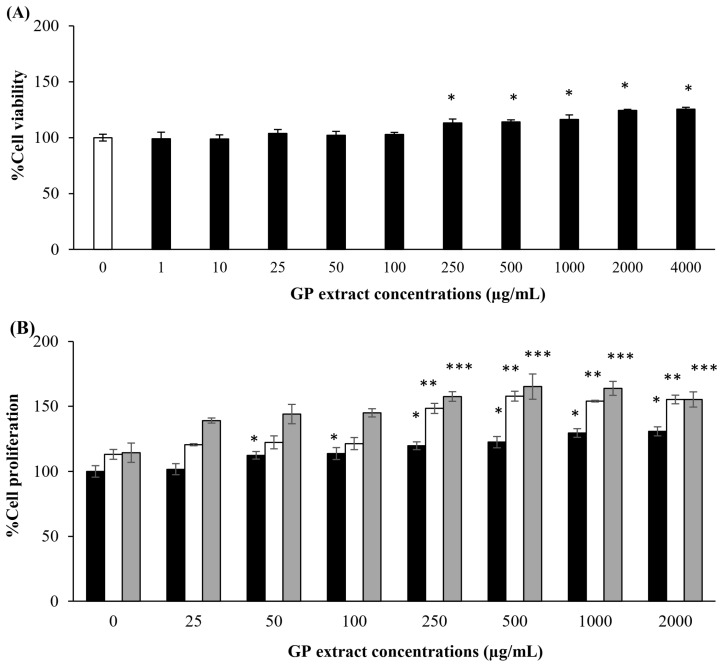
The percentages of viability (**A**) and proliferation (**B**) of HFDP cells following treatment with different concentrations of GP extract. The data are presented as the mean ± S.D. (N = 3). The * indicates a significant difference from the control group (untreated cells) at 24 h (*p* < 0.05), ** indicates a significant difference from the control group (untreated cells) at 48 h (*p* < 0.05), and *** indicates a significant difference from the control group (untreated cells) at 72 h (*p* < 0.05).

**Figure 2 pharmaceuticals-17-01084-f002:**
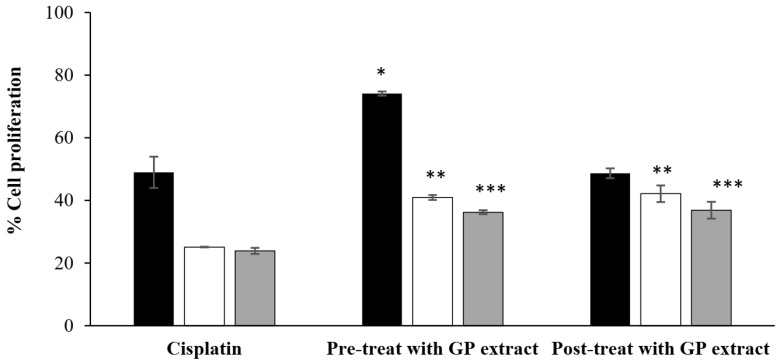
The percentage of HFDP-induced cell proliferation in the presence of cisplatin (30 µg/mL)-induced hair cell damage. Cells pretreated with GP extract before cisplatin treatment and posttreatment with GP extract after cisplatin treatment are shown. The data are presented as the mean ± S.D. (N = 3). The *, ** and *** indicate significant differences from the cisplatin-treated cells at 24 h (■), 48 h (☐), and 72 h (■), respectively (*p* < 0.05).

**Figure 3 pharmaceuticals-17-01084-f003:**
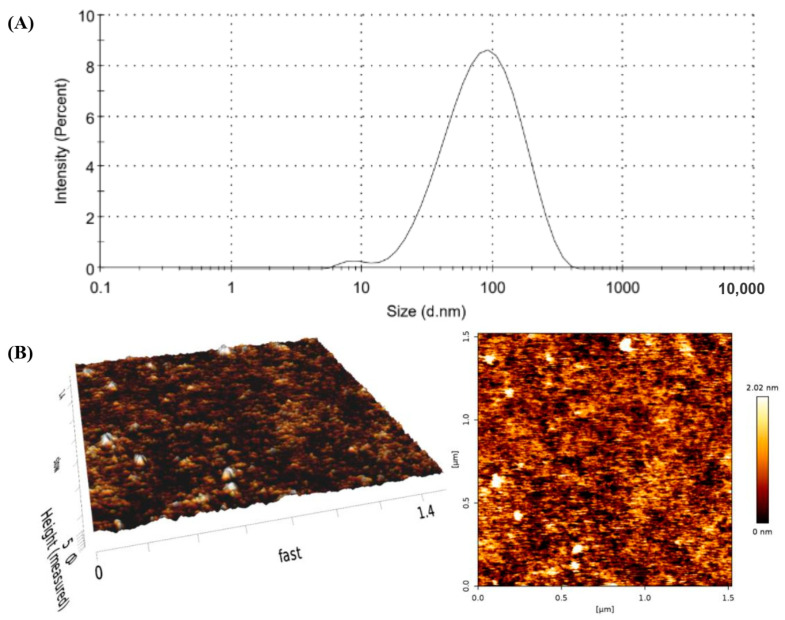
(**A**) Size measured by Zetasizer Nano-ZS and (**B**) AFM images of PL-loading GP extract.

**Figure 4 pharmaceuticals-17-01084-f004:**
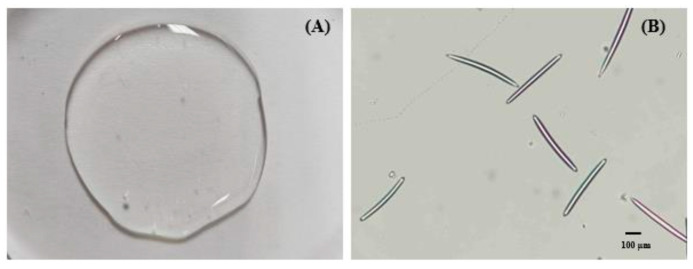
(**A**) Visual appearance and (**B**) microscopy images of GP extract-loaded PL–MS.

**Figure 5 pharmaceuticals-17-01084-f005:**
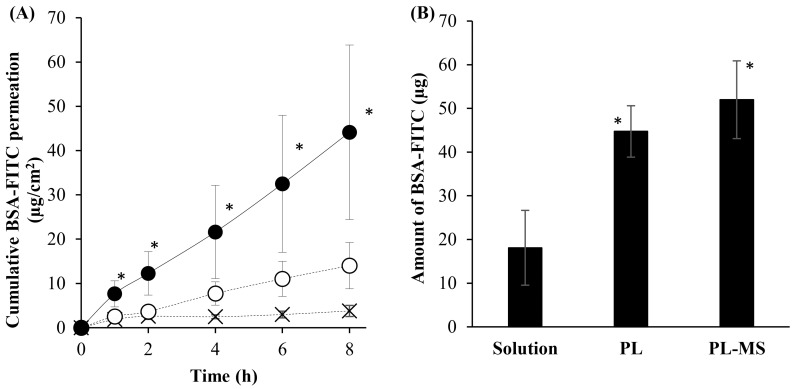
(**A**) Cumulative permeation of BSA-FITC versus time profiles of PL (○), PL-MS (●), and solution (×). (**B**) Amount of BSA-FITC deposited into the skin at 8 h of skin permeation. The * indicates a significant difference from the solution. The data present the mean ± S.D. (N = 3).

**Figure 6 pharmaceuticals-17-01084-f006:**
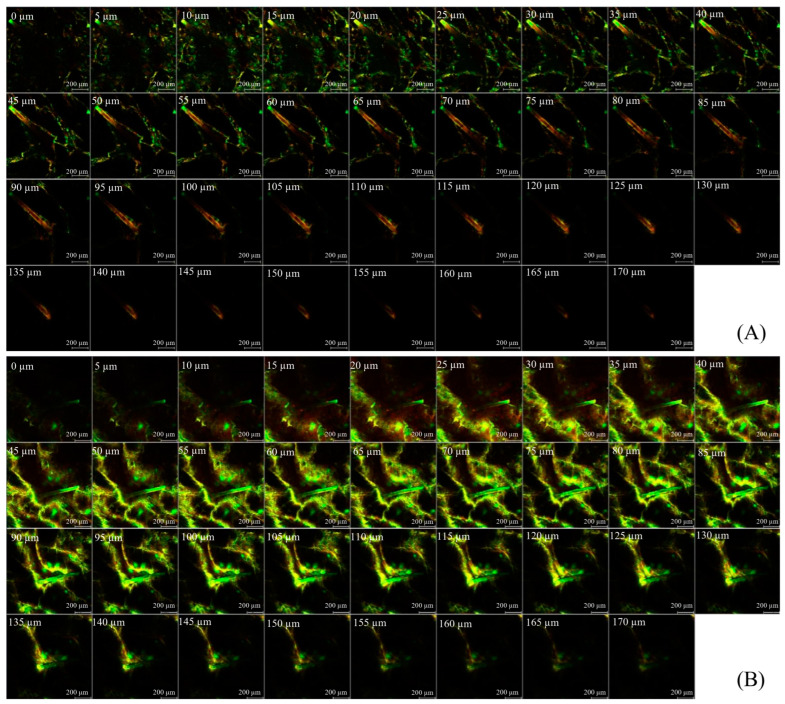
CLSM images (X-Y series) of skins treated for 8 h with BSA-FITC loaded PL (**A**) and PL-MS (**B**) (10× objective lens). BSA-FITC shows green fluorescence, and Rh-PE shows red fluorescence.

**Figure 7 pharmaceuticals-17-01084-f007:**
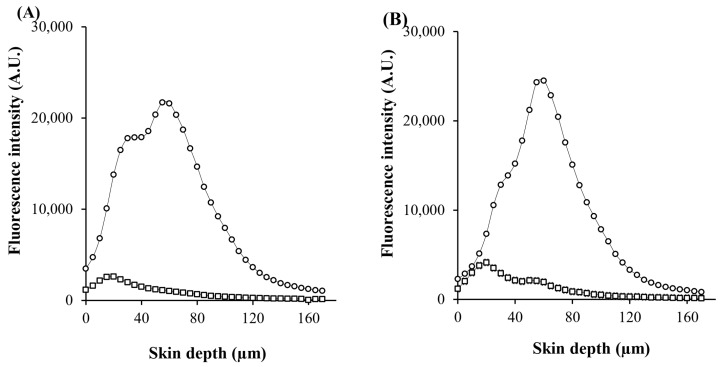
Fluorescence-intensity profiles of (**A**) Rh-PE-probed nanocarriers and (**B**) BSA-FITC-loaded PL (□), and PL-MS (◦) at different skin depths.

**Figure 8 pharmaceuticals-17-01084-f008:**
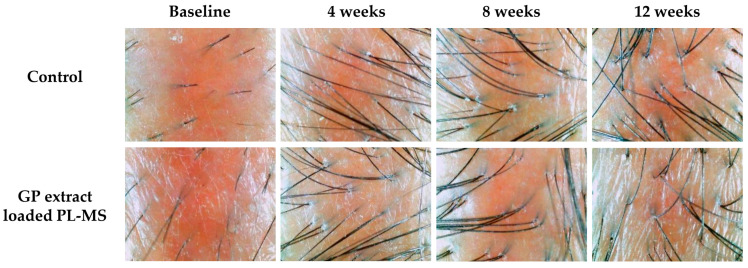
Images of scalp status (non-treated group; control) and its status after applying GP extract-loaded PL-MS at baseline (day 0), 4 weeks, 8 weeks, and 12 weeks.

**Figure 9 pharmaceuticals-17-01084-f009:**
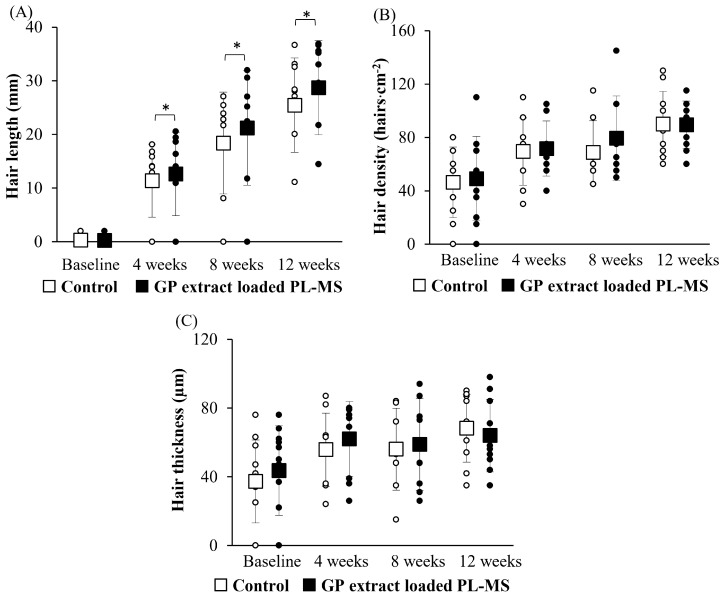
(**A**) Hair length, (**B**) density, and (**C**) hair-shaft thickness in chemotherapy-induced alopecia after applying topical GP extract-loaded PL-MS at baseline (day 0), 4 weeks, 8 weeks, and 12 weeks. Symbols: ○ and ● present the value of each volunteer in the control and GP-extract-loaded PL-MS-treated sites, respectively; and ☐ and ■ present mean ± S.D. of all volunteers in the control and GP extract-loaded PL-MS-treated sites, respectively. The * denotes a statistically significant difference from the control group (*p* < 0.05).

**Table 1 pharmaceuticals-17-01084-t001:** Skin-permeation parameters of all formulations.

Formulations	Flux (µg·cm^−2^·h^−1^)	K_p_ (×10^4^) (cm·h^−1^)	ER
Solution	0.36 ± 0.16	3.62 ± 1.58	-
PL	1.74 ± 0.68	17.40 ± 6.78	4.81
PL-MS	5.33 ± 2.49 *	53.29 ± 24.90	14.72

The * indicates a significant difference from the solution. Data present mean ± S.D. (N = 3).

## Data Availability

The original contributions presented in the study are included in the article/Appendix A, further inquiries can be directed to the corresponding author/s.

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
