# Peer review of "Novel Synergistic Approach for Bioactive Macromolecules: Evaluating the Efficacy of Goat Placenta Extract in PEGylated Liposomes and Microspicules for Chemotherapy-Induced Hair Loss"

_pharmaceuticals, 2024, doi:10.3390/ph17081084_

Round 1

Reviewer 1 Report

Comments and Suggestions for Authors

The authors identified and studied the efficiency of goat placenta extract in PEGylated liposomes (PL) and microspicules (MS) for chemotherapy-induced hair loss. The authors showed great results in terms of application. However, none can be found if we talk about the characterization of liposomes. Here are my comments:

1. The liposome images either in pristine or loaded must be investigated with an electron microscope. This result is necessary because the size liposomes that are usually prepared from thin film hydration would have sizes around the micron range without any extrusion follow-up methods. Please elaborate on this.

2. We also suggest the authors investigate the liposomes and the microspicule by using fluorescent microscopy.

3. Most cells are defined as a slightly negative charge (anionic) and this PL-MS is also described as having negative zeta potential. How can it work since at the molecular level, they would be repulsing each other?

4. Did the authors investigate the effect of the system (control experiment), without MS and Liposomes (only gel), Gel + MS (without PL), Gel + PL (without MS), and gel with both (the presented results). 

Comments on the Quality of English Language

1. The title is quite ambiguous; the meaning can also be interpreted as the system will induce hair loss. Please be careful with the context. 

2. Symbols in the Figure caption are also confusing.

3. The unit, please be consistent with either cm/h or cm.h-1

Reviewer 2 Report

Comments and Suggestions for Authors

Dear Authors,

Thank you for the opportunity to review your manuscript. I have identified several issues that require your attention:

  1. References: Please ensure all references are formatted according to the journal's requirements, both in the manuscript and the references list.

  2. Overuse of Self-Citation: The manuscript currently exhibits an excessive number of self-citations. Please reduce these citations to align with the journal's guidelines.

  3. Figure 1: The units on the X-axis should be uniformly formatted.

  4. Figure 2: The data for the CisPt group, particularly concerning the lack of standard deviation (SD), requires verification. Additionally, please provide an explanation for the absence of significant survival differences between the 48 and 72-hour groups, especially in cells treated with CisPt.

  5. Section 2.3: Include images depicting the size and zeta potential results, as analyzed by Zeta size software, and group these with the images in Figure 3.

  6. Figure 7: If possible, provide data for the placebo (non-treated) group on the same timeline as the PL-MS-treated group.

  7. Figure 6: The method used for fluorescence measurement should be described - whole intensity of the image or intensity of scpecific area.

  8. Figure 8C: Correct the Y-axis units, as they should be in millimeters (mm).

Future Recommendations:

  1. Consider including a positive control in in vitro, in vivo, and human studies.

  2. Provide images of a larger area of patients' scalps for all participants, to be included in the supplementary materials.

  3. There is a notable discrepancy in serum albumin content among the GP, DP, and PP groups. This discrepancy suggests potential issues with the treatment of biological tissues, possibly leading to protein degradation. Please verify these results and ensure that the treatment methods are appropriate and validated.

Comments on the Quality of English Language

Overall, the English language usage is well. However, there is an overuse of passive voice and occasional misuse of tense, particularly in the first half of the manuscript.

Reviewer 3 Report

Comments and Suggestions for Authors

The authors have evaluated the bioactivity of goat placenta (GP) extract on hair cells (both normal and chemotherapy-induced). They have attempted to  to develop PEGylated liposomes (PL) and microspicules (MS) formulations for promoting hair growth in patients with chemotherapy-induced hair loss. It's a good topic as hair loss due to chemotherapy is a universal phenomenon. They assessed the bioactivities of GP extract on human follicle dermal papilla (HFDP) cells and those cells that were damaged due to the effects of chemotherapy. 

GP extract was found to stimulate HFDP cell proliferation in both normal and cisplatin-damaged cells. Regarding methodology, GP extract was incorporated into PLs and MS gel (PL-MS), which were then investigated in vitro skin permeation and in vivo studies on the scalps of patients with chemotherapy-induced hair loss. The 12 weeks of therapy is reported to enhance the skin irritation-free hair growth rate significantly.

This remedy associated with chemotherapy is a good medical strategy having immediate clinical application possibility.

The article is well written. The methods section provides all in detail. Results have been presented professionally and supported with the conclusions. Fig. 8 demonstrates the concluding results on hair growth.

I also read the discussion and conclusion carefully and found no discrepancies. 

This article deserves publication.

Round 2

Reviewer 1 Report

Comments and Suggestions for Authors

Thank you for answering my question.

Reviewer 2 Report

Comments and Suggestions for Authors

The authors addressed all the issues I have mentioned.